# The Role of Intra-Patient Variability of Tacrolimus Drug Concentrations in Solid Organ Transplantation: A Focus on Liver, Heart, Lung and Pancreas

**DOI:** 10.3390/pharmaceutics14020379

**Published:** 2022-02-08

**Authors:** Gwendal Coste, Florian Lemaitre

**Affiliations:** 1Irset (Institut de Recherche en Santé, Environnement et Travail)—UMR S 1085, EHESP, Inserm, CHU Rennes, Université Rennes 1, F-35000 Rennes, France; gwendal.coste@univ-rennes1.fr; 2INSERM, Centre d’Investigation Clinique 1414, F-35000 Rennes, France

**Keywords:** intra-patient variability, drug, monitoring, pharmacokinetics, tacrolimus, liver transplantation, heart transplantation, lung transplantation

## Abstract

Tacrolimus, the keystone immunosuppressive drug administered after solid organ transplantation, presents a narrow therapeutic index and wide inter- and intra-patient pharmacokinetic variability (IPV). The latter has been fairly studied in kidney transplantation, where it could impact outcomes. However, literature about other transplanted organ recipients remains inconclusive. This review aimed at summarizing the evidence about the IPV of tacrolimus concentrations outside of the scope of kidney transplantation. First, factors influencing IPV will be presented. Then, the potential of IPV as a biomarker predictive of graft outcomes will be discussed in liver, heart, lung and pancreas transplantation. Lastly, strategies to reduce IPV will be reviewed, with the ultimate objective being ready-to-implement solutions in clinical practice by transplantation professionals.

## 1. Introduction

In recent years, intra-patient variability (IPV) of immunosuppressive (IS) drugs concentrations has surfaced as a potential novel biomarker in solid organ transplant recipients’ management. IPV is supposed to describe the variation of IS drug concentrations according to time (usually weeks or months) for a given patient, as opposed to inter-patient variability, which describes the variation of concentrations between patients. Several approaches have been proposed to evaluate IPV, but the most common is the simple calculation of a coefficient of variation (CV) of IS drugs trough concentrations (*C_min_*) (i.e., by dividing standard deviations of the concentrations by its mean). Less frequently is the standard deviation (SD) or the time in therapeutic range (TTR) used. Figure 1 graphically summarizes the parameters used to assess IPV. With tacrolimus (TAC), patients with higher IPV might have worse outcomes when considering inflammation, graft rejection, and finally, patient and graft survival. The mechanism behind this process may be related to the alternation of over and under-subclinical immunosuppression episodes. These episodes could lead to drug toxicity in the first case and to immune activation in the latter case. As a consequence, the graft can accumulate lesions progressively conducting to its dysfunction. Most of the studies published on that topic so far refer to kidney transplantation, but it has also been shown that IPV might impact other solid organ transplantation (SOT). The aim of the present review is then to summarize the information on TAC IPV and its consequence in SOT, excluding kidney transplantation. A special focus on factors influencing IPV and potential interventions aiming at reducing this IPV will also be reviewed.

## 2. Factors Influencing IPV

Of course, factors influencing IPV include pre-analytical and analytical variability, which is virtually affecting every bioanalytical assay. As the gold standard of the TAC *C_min_* determination is the high performance liquid chromatography coupled with tandem mass spectrometry, a method widely used in developed countries, imprecision in IPV can, therefore, be considered as relatively low. The more important dosing flaw in TAC *C_min_* determination is the error in dosing timing, i.e., the trough concentration not being a real one. This error is now quite well recognized. Despite the bioanalytical issue being of interest, we will here focus on biological factors influencing IPV. Most of the studies conducted on that topic considered IPV in stable patients, which are patients with at least three months since the transplantation. As patients are far from the acute post-operative phase, IPV is thought to be mainly related to adherence at that time. During the early post-operative period, however, other factors can account for IS drug variability, and it might be an elective period for interventions aiming at acting on IPV [1]. Among factors influencing IPV, genetics might explain part of the TAC *C_min_* variability. However, for the moment, the *CYP3A5* genotype has not been definitely recognized as associated with TAC variability. Nevertheless, IPV has always been evaluated after three to six months in the studies conducted on that field [2,3,4]. In contrast, it has been widely reported that transplant recipients with higher IS drugs clearance associated with the *CYP3A5*1* genotype may require a longer time to reach the concentration target during the early treatment phase [5]. Eventually, the analysis of a large observational study conducted in 1472 kidney recipients during the first six months post-transplantation shows that TAC IPV decreases with the increase in loss-of-function alleles. Each additional *CYP3A5* loss-of-function allele (*3, *6 or *7) decreases patients CV by 1.61% for African American (AA) patients and by 1.85% for European ancestry patients, although the association was only statistically significant in the latter case, probably due to a low power in the AA group (*p* = 0.07 and *p* = 0.0042, respectively) [6]. The mechanism behind these findings could be a greater elimination rate in *CYP3A5*1* carriers, leading to more rapid variation of TAC concentration, and/or a greater susceptibility to drug/drug interactions because of an increased metabolic activity. Other factors can also contribute to pharmacokinetic variations such as drug administration regarding food intake, drug-drug and alimentary interactions, gastro-intestinal disorder, and chronopharmacology [7]. Hematocrit and bilirubin have been shown to be predictors of TAC IPV during post-transplant year-three in a pediatric cohort of patients [8]. The impact of drug formulations on variability has also been studied. Hence, TAC can be administered through three different formulations: immediate-release, prolonged-release, and a meltdose formulation. This part will be further developed in the interventions aiming at reducing the IPV section.

Thus, several biological and clinical factors seem to significantly influence IPV of TAC *C_min_*. Their impact on IPV will also be discussed in this review.

## 3. IPV in Liver Transplantation and Outcomes

The IPV of TAC *C_min_* has been studied in adult and pediatric liver transplantation (LTx). The first study conducted on that topic has been made by Venkat and colleagues, who retrospectively analyzed 101 pediatric LTx recipients treated by TAC for at least one year. High SD (considered as a continuous variable) was associated with late acute rejection (AR) in the multivariate regression. One unit increase in SD was associated with an odds ratio of 3.49 (95% CI = 1.31–9.29) of having a late AR. Ten episodes of late AR occurred in 11 patients with SD greater than 2 ng/mL [9]. Several other teams tried to replicate this study in pediatric liver patients. Hence, Stuber et al. showed, using 68 pediatric LTx recipients files, that a prespecified SD greater than 3 ng/mL was significantly associated with biopsy-proven acute rejection (BPAR) (OR = 4.0). Patients had to have one transplant only, be at least one year after the graft, and have at least three TAC *C_min_* available. Half of these patients were under four years of age at transplant. A receiver operating characteristics (ROC) curve approach identified a cut-off SD of 2.5 ng/mL as the best predictor for BPAR [10]. The SD calculated over at least three TAC *C_min_*, referred to as the “medication level variability index” (MLVI) by the group of authors, was used by Shemesh and colleagues in a multi-center prospective study on 379 pediatric LT recipients followed for two years [11]. Pediatric LT recipients at least one year after their surgical procedure were included. The objective was then again to study the impact of MLVI on graft rejection, and to determine the best cut-off value. Mean pre-rejection SD was significantly higher in patients who had late AR (2.4 ng/mL vs. 1.6 ng/mL, *p* = 0.003). Continuum as well as dichotomous SD (threshold of 2.5 ng/mL) predicted rejection. The ROC curve analysis found MLVI as a significant predictor of rejection but with a low AUC (0.61). The best cut-off value was found to be 2 ng/mL for the entire cohort, and 2.5 ng/mL for the adolescent sub-group. De Oliveira et al. [12] reported 50 pediatric TAC-treated LT recipient cohort outcomes, focusing on children younger than 12-years of age. Patients were transplanted between 1999 and 2011 and were included at least 13 months after the graft implantation. Data related to the second and third post-transplant year. Patients had to have at least five serum TAC *C_min_* measurements, excluding inpatients measurements and those with a contemporary drug-drug interaction toward TAC. In this study, no correlation was found regarding rejection. Alanine-amino-transferase elevation was significantly more frequent in patients with a SD greater than 2 ng/mL. This relatively small cohort with few observed rejection episodes (13 in 13 patients) failed to reproduce the observations of the previous studies.

Regarding adult patients, Lieber and colleagues explored two retrospective liver transplant cohorts between 6 and 18 months after transplantation, looking for non-adherence markers. The authors showed, in the exploration cohort, that the SD was associated with clinician and self-reported adherence. An increase in SD was associated with an increased risk of graft failure in the independent cohort (1.005 per unit increase in SD, *p* = 0.04) [13]. Christina et al. also studied the medication level variability index (MLVI), another name for SD. They retrospectively randomly included 150 LT recipients from their database who were transplanted between 1988 and 2010, who received TAC between 2007 and 2010 for at least six months and had at least three *C_min_* measurements. Patients with BPAR had higher SD than patients without rejection (3.8 ± 3.2 versus 2.6 ± 1.6 ng/mL). Rejectors had significantly more chances to have a SD above 2.5 ng/mL than non-rejector patients. Patients with one rejection and patients with several had comparable MLVI. The ROC curve analysis revealed that a cut-off of 2.0 ng/mL brought 77% sensitivity and 60% specificity, and a threshold of 1.8 ng/mL had 92% sensitivity and 48% specificity to predict rejection. One of the study’s limitations is the inclusion period, which is very large, calling into question the transposability to modern immunosuppression strategies [14]. A study conducted by our team [15] introduces the notion of IPV calculated as CV (patient SD divided by patient mean concentration) for the first time in LTx. Hence, two patients may have the same SD, but may not have the same CV given the fact they may have different mean concentrations. In our study, IPV was calculated during the first post-transplantation month (excluding the first week) in de novo adult LT recipients transplanted between January 2002 and December 2014, excluding patients who died during the first two weeks. Patients were divided in two groups according to a cut-off value of 40%, corresponding approximately to the 3rd quartile. The analysis group included 812 LT recipients. The median follow-up was 61.8 months. The median CV was 31.4% (IQR = [23.5%; 41.8%]). The MELD score and Child-Pugh grade were associated risk factors for a high CV. AR rates were similar between groups but the high CV (>40%) group experienced significantly more neurological, cardiovascular and renal complications as well as a longer hospital stay than the low-CV group, whereas three-months graft survival was similar between groups, and one-year and long-term graft survival was significantly lower in the high CV group. In the multivariate analysis, CV > 40%, HCV infection, and liver malignancy were associated with graft loss. Even if causation cannot be firmly stated, the hypothesis of an alternation of overexposure and underexposure periods resulting in organ damage (toxic or immunologic lesions), in high CV patients, might explain the worse outcomes. Given the very early period of IPV assessment where patients are strictly supervised for treatment intake, variability seems more related to individual pharmacokinetic factors than adherence in this study [15]. The same year, Del Bello et al. reported the results of a study with 116 adult LTx recipients with a functioning graft at one month, between February 2008 and June 2015. Patients had at least 3 TAC *C_min_* (median = 10). As patients did not receive the same dose during all follow-up, dose-corrected IPV was also calculated. IPV evaluated as a CV was 32 ± 12% (median = 30.5). Twenty-two patients had at least one rejection episode. In a multivariate analysis, TAC *C_min_* below 5 ng/mL, CV as a continuous variable, CV > 35% and CV > 40% were predictors of BPAR. All those factors, excepted low TAC *C_min_* were also predictors of de novo DSA detection. In both cases, CV > 40% had a better predictive value. IPV was not related to death (*n* = 6) or CMV replication (*n* = 24) possibly due to the limited number of patients included [16]. All the previous observations are in contrast with those reported by van der Veer and colleagues. They evaluated IPV starting from six months after transplantation, where IPV is thought to be more related to adherence. The primary outcome was a composite endpoint of immune-mediated graft injury and late acute cellular or chronic rejection after month-6. IPV was calculated as a CV in patients with at least five *C_min_* measurements between month-six and -18. The study retrospectively included 326 adult LT recipients, treated with TAC for at least 18 months. Patients were divided into two groups according to median CV. The median follow-up time was 5.2 years. Considering the median CV of 28% as a cut-off point, the author did not show a more frequent primary endpoint in the most variable patients, even if the survival curves showed good separation (*p* = 0.068). IPV, considered as a continuous variable, finally failed being a risk factor for primary outcome in the multivariate analysis, and only the MELD score and rejection within the first six months were associated with the primary outcome. However, a high CV was associated with a loss of renal function in patients with an eGFR below 40 mL/min/1.73 m^2^ at six months.

Despite one study in children and one in adults with opposite results, pediatric and adult liver transplant recipients presenting high IPV, starting from early post-operative days, seems to have worse outcomes, and TAC variability should be addressed in this population of patients. In LT recipients, IPV causing under-exposure could lead to immune micro-aggression of the transplanted liver, reducing its metabolic activity, which is almost the only driver of TAC elimination. It could at last be the cause of a toxic TAC accumulation.

## 4. IPV in Heart Transplantation

Apart from preliminary reports in scientific meetings, few papers reported IPV relevance in heart transplantation before 2018. Only Pollock-Barziv and colleagues reported results for a small subset of heart transplant (HT) patients in their 2010 retrospective study. In this study, 14 pediatric patients with an acute cellular rejection at least six months post-transplant, treated with steroids or by an increase in immunosuppressive drug dosage, were compared to 14 controls. SD was measured in the immediate months before rejection or the last six months of patient follow-up. Pooled results with other type of transplantations showed an increased rate of rejection and graft loss in patients with SD above 2 ng/mL [17]. These results have been very recently confirmed in a retrospective study including 118 pediatric HT recipients. Sirota and colleagues calculated SD from one-year post-transplantation, and showed a numerical increase in poor outcomes in patients with SD above 3 ng/mL, considering rejection (73.4 vs. 46.3%) and death (20.3 vs. 5.6%). SD from one to five years post-transplantation, considered as a continuous variable, was associated with the onset of a composite endpoint combining cardiac allograft vasculopathy (CAV), retransplantation, and death [18].

In adult patients, Gueta et al. retrospectively analyzed TAC *C_min_* CV in 72 TAC-treated HT recipients from month-3 to -12. Primary endpoints were BPAR and death. The median TAC *C_min_* CV was 28.8%, and mean *C_min_* did not differ between groups. There was no association between the CV > median and rejection evaluated on endomyocardial biopsies up to one year post-transplantation. However, the high CV group had an 8.52 risk increase of having any rejection after one year [19]. This study again raised the interest of an early evaluation of IPV in solid organ transplantation, which might be related to altered pharmacokinetics, rather than adherence [1]. In contrast, Shuker et al. reported, in an 86 HT recipient cohort co-treated with rATG induction, MMF and CS, the lack of a relationship between IPV of TAC *C_min_* and the progression of CAV between two coronary angiographies performed at one and four years post-transplantation, and acute rejection. It is noteworthy that IPV was evaluated after six months, and that the median IPV was low (17.7%), suggesting a very high adherence in the population study [20]. A less commonly explored IPV marker, the TTR has been recently evaluated by Baker and colleagues. In this retrospective 67-patient cohort of HT recipients, the authors assessed the time during which the TAC *C_min_* remained in the range of 10–15 ng/mL (assuming a linear relationship between two concentration measurements). Of note, in their center patients do not usually receive induction treatment (except a methylprednisolone bolus), and only patients treated without basiliximab have been included. The primary endpoint was the TTR from day-0 to day-30, in patients presenting acute rejection or not. TTR was not different between the groups (31.4 vs. 36.12%) in the rejecting and non-rejecting group, respectively. A major limitation of this study was to include the first post-operative days where the TAC *C_min_* are still unstable, which might induce a bias in evaluating IPV [21].

While limited data exist, IPV may have an impact on patients’ outcome in the HT era. As heart blood flow drives the entire body blood flow, and as TAC AE include toxicity directed toward the heart (high blood pressure, diabetes…), IPV could largely influenced heart condition, including CAV (chronic rejection) in HT.

## 5. IPV in Other Transplantations

In 2005, Knoop et al. briefly reported data on TAC IPV in lung transplant recipients. In a three time points non-compartmental pharmacokinetic study in 22 lung transplant recipients (half of them for cystic fibrosis indication) at steady-state, cystic fibrosis and non-cystic fibrosis patients displayed a comparable and relatively low TAC *C_min_* IPV (15.7% vs. 13.4% respectively, *p* = n.s.). Thus, in stable lung transplant recipients, IPV was found to be modest [22]. Gallagher and colleagues evaluated IPV in a 110-lung transplant recipient’s cohort. They studied the links between individual TAC *C_min_* SD (calculated during 3 distinct periods: 0–6 months, 6–12 months and 12–24 months) and chronic lung allograft dysfunction (CLAD) or death, during a median follow-up of 60 months. SD decreased from period 1 to period 3. TAC *C_min_* SD was not associated with CLAD or death during the period 0–6 months, while during months 6 to 12, for each SD unit increase, the risk of CLAD and death increased by 46% and 27%, respectively. Moreover, the mean TAC *C_min_* was negatively associated with CLAD. In contrast, no predictor of AR was found in this study [23]. Ensor et al. evaluated TTR in lung transplant recipients in a single-center retrospective study. Lung transplant recipients were included if they remained on a TAC-based immunosuppressive regimen during the first post-transplant year. Induction treatment was alemtuzumab or basiliximab, and maintenance treatment associated TAC, mycophenolic acid and prednisone. Protocol biopsies were performed every 2–3 months, and TTR was calculated at one year using Rosendaal’s method [24]. In the multivariate analysis, increasing TTR by 10% reduced the risk of any rejection by 19%, of high-grade rejection by 36%, of CLAD by 40% and of mortality by 52%. Increasing TAC *C_min_* also decreased the risk of rejection, but only for high-grade ACR. In this study, TTR appears superior to other IPV markers (SD and CV), regarding association with rejection [25]. Finally, Kao et al. explored the relationship between ACR and IPV (expressed as CV or TTR), measured from day one up to six months. In this study, no relationship was evidenced between patients with (*n* = 73) or without rejection (*n* = 84), even when considering rejections with an ACR ≥ 2, or when splitting the population of patients in lymphocytic bronchiolitis/no lymphocytic bronchiolitis [26].

Only two studies explicitly reported the importance of TAC IPV in pancreas transplant recipients. In the first one, Torabi et al. retrospectively analyzed 39 simultaneous kidney/pancreas transplant recipients continuing on immediate-release (IR) TAC (*n* = 21) or converted to extended-release TAC (LCP-Tacro) (*n* = 18). Primary endpoints were TAC *C_min_*, IPV as a CV and BPAR. TAC *C_min_* IPV was comparable at one, three and six months between groups. A greater difference was observed at 12 months (IR: 41.0% vs. LCPT: 33.1%; *p* = 0.18), but this was not statistically significant. While no difference was observed in creatinine levels between groups, glycemic control assessed by HbA1c levels was statistically better at three, six and 12 months in the LCPT group. There were six BPAR in the IR group and none in the LCPT group (*p* = 0.03). In this study, the differences observed may be more related to the pharmacokinetics properties of the drug formulation than to IPV (which is not significantly different). However, the once-daily formulation could help diminishing IPV by improving adherence through drug burden decrease [27]. Besides, Davis et al. reported an observational study including 538 kidney and simultaneous kidney/pancreas transplant recipients without any specific information on the number in the latter group. They found worse graft outcomes, in terms of dnDSA, and death-censored graft loss by five years in patients with high IPV (TTR < 40% and CV > 44% as identified by ROC curve analysis). Considering IPV markers as continuous variables, only TTR was associated with pejorative outcomes. Finally, patients combining a high CV and a low TTR had the highest rate of adverse events, whether it is 12-month dnDSA, 12-month rejection or five-year death (54.0%, 82.7% and 81.4%, respectively) [28]. Again, the particular impact on the kidney/pancreas is largely unknown given the lack of subgroup analysis in this study.

In other organ transplantations, despite the limited studies conducted, high IPV seems to affect patient outcomes with the largest amount of data available for TTR.

## 6. Strategies to Reduce IPV

Given the identification of some risk factors contributing to IPV, different approaches have been proposed to reduce TAC variability. Again, most of the studies in this field have been performed in kidney transplantation and a number of studies related to kidney transplant recipients will be reported in this part to illustrate important points.

### 6.1. Switch to a Once-Daily Formulation

Data comparing IPV giving pharmaceutical formulations should be interpreted according to the period of transplantation, the level of drug adherence of the group and the type of formulation. Hence, Leino and colleagues reported a lack of difference in seven-day IPV in a cross-over study with the drug innovator and two other generic drugs. This study included twenty-five liver and twenty-five kidney stable (>6 months) transplant recipients. Patients where highly adherent, as confirmed by the low CV (15.2%) in the population, but CV did not differ between formulations [29]. In contrast, when using area under the curve of TAC concentrations CV, Stifft and colleagues showed a decrease in CV in 40 kidney patients converted to a once-daily formulation. However, when considering TAC *C_min_*, the difference vanished and the results seemed mainly driven by the 11 patients with a *CYP3A5*1/*3* genotype [30]. After six months, the potential impact of converting patients to a once-daily formulation is thought to be related to an increase in patients’ adherence by reducing the burden of dose administration. There are insufficient data to hypothesize a role of once-daily conversion in the early post-operative period.

Considine et al. retrospectively analyzed 129 adults and adolescents (*n* = 15) benefiting from a liver transplantation between 1989 and 2012 that were switched from IR-TAC to extended-release (ER) TAC between 2007 and 2012. They were compared to 60 control patients that remained on IR-TAC that were transplanted between 2009 and 2011. Converted patients were divided into two groups of early conversion (occurring in the first month after the transplantation, *n* = 64) and late conversion (after the first month, *n* = 65, most of them being converted more than six months after transplantation). IPV was calculated using only SD, one week after Tx for the IR-TAC group, and starting from the conversion for the ER-TAC groups. Of note, the patients in the late conversion group were largely younger at transplant compared with the two other groups. The early conversion group displayed the lower IPV, while in the late conversion group, patients showed a significant lower SD of TAC *C_min_* after conversion. However, the lack of comparability of the three groups, the retrospective design and the different timing of evaluation between the groups makes it very hard to conclude on IPV [31]. In a large cohort (*n* = 345) of solid organ recipients, Del Bello et al. identified 54 patients (30 kidney, 20 liver, three kidney-liver and one kidney-pancreas recipient) with at least three TAC *C_min_* before and after a switch from IR-TAC to ER-TAC after a mean treatment duration of 23.5 months. They reported no difference in CV before (29%) and after (24%) the swich (*p* = 0.65) [32]. Finally, a cross-over bioequivalence study of IR-TAC vs. two generics performed on 36 liver and 35 kidney transplant recipients by Alloway et al. found no difference in IPV of AUC between the tested medications. Moreover, the *CYP3A5* and *ABCB1* genotype did not influence IPV [33].

### 6.2. Implementing Patient-Based Interventions

Patient-based interventions can be used to reduce IPV in transplant recipients. They used either trained specialists (pharmacologists, transplant pharmacists) or communication systems information and communication technology. Back in 2008, Shemesh and colleagues implemented a clinical program aiming to improve adherence among pediatric LT recipients. This proof of concept study targeting non-adherent patients explored the feasibility and pertinence of common sense adherence-positive actions in a 23 pediatric LT recipients’ cohort. Adherence was assessed using TAC *C_min_* SD considering an SD above 3 ng/mL as a marker of non-adherence. After program implementation on the six non adherent patients, three became adherent with an SD decreasing below 3 ng/mL. The program consisted mainly in an increase in the frequency of outpatient visits and TDM as well as education actions. Costs did not increase throughout the study, despite a protocol-mandatory increase in outpatient visits [34]. The same team recently reported interventions in 15 pediatric lung transplant patients. The program consisted in a remote contact with patients aiming at identifying barriers to adherence and reducing the frequency of inadequate drug intake. Fifteen patients included in the study and benefiting from the intervention were compared to seven controls. The majority (75%) of patients completed the three phone calls of the study. In the intervention arm, the SD was significantly reduced at day-180 (−1.62; *p* = 0.02) compared with baseline. At day-180, the difference in SD between the intervention and control group approached significance (−2.21, *p* = 0.054).

To our knowledge, randomized control trials aiming at comparing the impact of patient-based interventions to standard management on IPV have only been conducted in kidney transplantation with some positive and negative results [35,36,37,38].

## 7. Conclusions

IPV thus appears as an interesting tool to monitor TAC therapy and, to a limited extent, predict its effects. Indeed, by the time the IPV is observed, the eventual harmful consequences might already be determined and hardly modifiable. However, this biomarker could predict long-term AE related to drug over- and under-exposure, and it could help identifying sub-populations that could benefit from an IPV-reducing intervention. Moreover, IPV is easy to implement into clinical practice, as it does not necessitate additional measures other than *C_min_*. However, several considerations have to be taken into account, particularly the method of calculation (SD, CV, TTR), and the time period post-transplantation assessed, as the factors influencing IPV may differ. Additionally, studies’ methodologies on the topic is heterogeneous, and mostly of a retrospective nature. As intercurrent clinical events are susceptible to lead to *C_min_* variations and thus IPV increase, the role of IPV as a cause or a consequence of worse outcomes is not definitely conclusive. The prospective evidence of a superiority in terms of outcomes, of reducing the IPV of TAC *C_min_* in transplanted patients, has still to be defined. Randomized control trials with clinically relevant endpoints should be encouraged to further evaluate the added value of this biomarker.

## Figures and Tables

**Figure 1 pharmaceutics-14-00379-f001:**
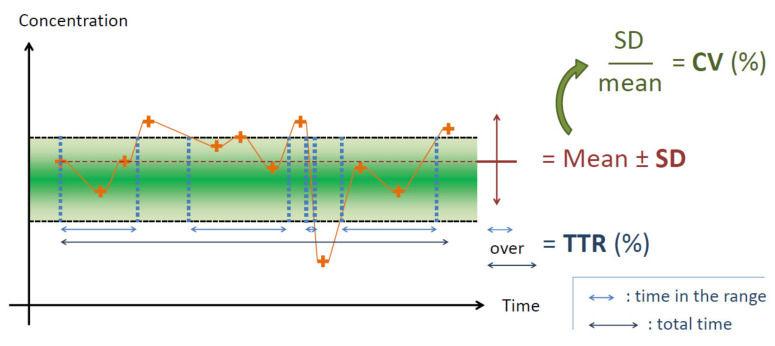
Overview of different calculation methods for IPV, highlighted in bold text. Orange crosses represent TAC concentrations, black dashed lines represent the therapeutic interval, light blue double arrows represent the time during which the patient is in the therapeutic interval assuming linear evolution of *C_min_* between occurrences, the dark blue double arrow represents the total measurement time. *C_min_*: whole blood tacrolimus trough concentration; CV: coefficient of variation; IPV: intra-patient variability; SD: standard deviation; TAC: tacrolimus; TTR: time in therapeutic range.

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
