# Peer review of "The Role of Intra-Patient Variability of Tacrolimus Drug Concentrations in Solid Organ Transplantation: A Focus on Liver, Heart, Lung and Pancreas"

_pharmaceutics, 2022, doi:10.3390/pharmaceutics14020379_

Round 1

Reviewer 1 Report

It is a novel study about the unmet needs of immunosuppressant use.

I have no comment on this manuscript

Author Response

We thank reviewer #1 for his positive comment on our work.

Reviewer 2 Report

This review carefully discusses the relationship between the intra-patient variability (IPV) of tacrolimus blood concentrations and the prognosis of solid organ transplant patients.

Although it is titled “Outside of kidney transplantation”, it effectively includes data from kidney transplant patients to explain the position and concept of IPV of tacrolimus blood concentrations.

The findings on standard deviation (SD), coefficient of variation (CV), and time in therapeutic range (TTR) are also reviewed appropriately.

The findings in pediatric and adult patients are captured separately in this review, and the pharmacokinetics are accurately assessed.

In addition, the challenges and solutions for IPV of tacrolimus blood concentrations have been correctly reviewed.

However, in Conclusion section, the argument that IPV is a tool for predicting the effect of tacrolimus is a bit problematic and should be considered for revision.

I think it is correct that the degree of IPV determines the prognosis of a patient, so I think it is also a valid argument that reducing IPV leads to improved prognosis. On the other hand, in a patient, the effect of tacrolimus is a consequence of the degree of IPV. By the time the IPV of a patient is known, the prognosis of that patient has already been largely determined, and although it may be possible to predict, it may not be possible to treat. Therefore, I think it is a bit unreasonable to interpret it as a "prediction tool".

Epidemiological data showing the prognostic effect of IPV reduction may be useful for patient adherence education.

Author Response

See file attached

Reviewer 3 Report

I've read the manuscript by Coste and Lemaitre titled: "Outside of kidney transplantation: the role of intra-patient variability of tac drug concentrations in solid organ transplantation".

The authors review the literature on IPV across non-kidney Tx. They conclude, in broad terms, that IPV generally is a predictor of TAC effects, and thus represents an interesting tool, also because it is easy to implement.

First and foremost I am reluctant to agree with the authors wrt the general conclusion. I think more studies are needed in order to conclude anything. IPV is tricky as a biomarker because it essentially is a composite measure of tacrolimus ADME, practical particulars like sampling and analysis adherence including assay variability, and simple patient drug adherence. It is expected that nonadherence will lead to poor effects. It is not clear to this reviewer at least, whether biology contributes to IPV in a clinically significant way. To me, that is the important question. And I don't think the authors shed much light on this in the review. I suggest that the authors address these points directly in the introduction. Also, discuss how IPV is determined in practice where sampling can be done as part of a routine or due to current issues with concurrent illness.

Title
I suggest that the title be reconsidered such that kidney transplantation is replaced because the review covers all SOT except kidney. Also for search purposes.

Introduction
I wonder what the background IPV is, ie the variability caused by unprecise dose time recording, ditto for blood sampling, and assay variability. It would be nice to know the magnitude of this in order to better assess the unexplained variability (and/or adherence issues).

What if anything is known about factors of the estrogen cycle and impact on IPV? Seasonal variation? What is known about subpopulation contributions of sex? (women are less adherent than men according to pmid: 24206025) and race?

Genotype (CYP3A5 expression etc) is discussed. Please expand a bit on how genotype would contribute to IPV.

Is anything known about IPV of other drugs which are used together with Tac? Mycophenolate mofetil ? Glucocorticoids? Might knowledge on this be helpful?

The individual paragraphs
covering the different SOT types could be improved by adding a sentence or two on why the particular condition would itself contribute to IPV. What makes a HT different from Ltx. Other drug regimens?

Other comments
The language is acceptable but can be improved.

Line 68 - Consider replacing the word "lead" with "contribute"   
Line 82 - "on" - I don't quite understand.
Line 109 - remove s from "adults"
Line 215 - spelling error in "non-compartmental"
Line 261 - what does "free-adverse" mean?
Line 290 - "group" should be "groups"
Line 294 - "two times younger". What does that mean? "Half as old"?

Author Response

See file attached

Round 2

Reviewer 2 Report

In response to my suggestions, the manuscript has been appropriately revised.

I believe that this is a useful article for Pharmaceutics readers.

Reviewer 3 Report

The authors have addressed all the points I raised to my satisfaction. I have no further comments except for the sentence "could lead to immune micro-aggression of the transplanted liver...". I am not familiar with the term micro-aggression in regards to liver function. The fault is probably mine.